# Analysis of Axial Acceleration for the Detection of Rail Squats in High-Speed Railways

**Hojin Cho \*, Jaehak Park and Kyungsu Park**

302, 105, Shinil-dongro, Daedeok-gu, Daejeon 34326, Republic of Korea; jaehak@igsg.co.kr (J.P.);
kyungsu@igsg.co.kr (K.P.)
\*  Correspondence: hojin@igsg.co.kr

**Abstract:** A squat is a type of fatigue defect caused by short-wavelength rotational contact; if squats are detected early, the maintenance cost of the track can be effectively reduced. In this paper, a method for the early detection of squats is presented based on ABA (axle box acceleration) and frequency signal processing techniques. To increase the measurement sensitivity for the squat, ABA was used to measure the longitudinal vibration. Compared to vertical ABA, longitudinal ABA does not include vibrations from rail fasteners and sleepers, so it is possible to effectively measure the vibration signal in relation to the impact of the rail. In this paper, vibration data were measured and analyzed by installing a 3-axis accelerometer on the wheel axle of the KTX; squat signals were more effectively extracted using the longitudinal vibration measurement presented above. The algorithm to detect the position of squats was developed based on wavelet spectrum analysis. This study was verified for the section of a domestic high-speed line, and as a result of conducting field verification for this section, squats were detected with a hit rate of about 88.2%. The main locations where the squats occurred were the rail welds and the joint section, and it was confirmed that unsupported sleepers occurred at locations where the squats occurred in some sections.

**Keywords:** axle box acceleration (ABA); rail transportation maintenance; railway monitoring; surface defects on railway rails

## 1. Introduction

In the railway industry, a wide range of monitoring systems have recently been developed to facilitate manual railway inspection operations [1–5]. Among the most widely used methods for railway monitoring is ultrasonography [6]. This method, however, is only effective at detecting severe defects, such as those with a depth of 5 mm or more. There are other methods available, such as visual inspection [7], AC-field techniques [8], image recognition [9], the attachment of a strain gauge to a wheel pair [10], acoustic detection [11], and status monitoring based on stress wave detection [12]. These methods all have their own advantages and disadvantages in terms of cost-effectiveness, reliability, and accuracy. In this regard, the present study proposed a novel method of vibration measurement based on axle box acceleration (ABA). In previous studies focused on the application of accelerometers, accelerometers were mainly used for monitoring and control purposes [13–16]. It was reported that railway monitoring using an accelerometer was quite effective at detecting rail corrugation, defective rail welds, and degraded insulation joints. Compared to other status-monitoring systems, ABA is advantageous mainly in terms of the cost and ease of maintenance and also in that it can be implemented in trains that operate at commercial speeds. Table 1 summarizes the findings of previous studies on the application of ABA systems by country. ABA systems have been implemented for rail defect detection and analysis in various countries, including Korea [17], the Netherlands [18], Japan [19], Poland [20], and Italy [21].

**Table 1.** ABA systems in various countries.

| ABA Implementation | Defects Studied | Frequency Range | Railway Network |
|---|---|---|---|
| Vertical ABA, 2 sensors, 80 km/h | Corrugation, wavelengths 0.055–0.080 m | Measured up to 2864 Hz | Polish State Railways Network [20] |
| Vertical ABA Lateral ABA | Corrugation lateral discontinuity, curve rail wear, damaged switches | Analysis between 25 Hz and 1246 Hz | Subway of Milan, Italy [21,22] |
| Vertical ABA Lateral ABA | Alignment, zones of large lateral force, track irregularities below 100 Hz | Analysis up to 100 Hz | Japanese high speed railways [23] |
| Vertical ABA Lateral ABA speed 300 km/h | Long wavelength irregularities, from 3 m to 200 m, both lateral and vertical irregularities | Analysis up to 2048 Hz | Korean train Express, high-speed line [17] |

Keeping the rail in good condition requires systematic and periodic testing. A computer-based tool is used for such tests, and rail managers implement systematic maintenance to minimize the total cost and secure the long-term quality of the rail [22]. Various methods can be used to monitor rail track conditions [24–28].

A visual inspection alongside ultrasonic and eddy current measurement methods is used in Korea and many countries to detect short-wavelength defects in railways [29,30]. However, these types of tests are effective only in cases in which the rail performance decreases due to severe damage; they are not optimal methods for maintenance. In addition, a visual inspection requires a large amount of labor, and different results can be derived by different workers [7]. Therefore, it is necessary to develop a method that automatically detects irregular defects, such as squats, and a monitoring system to efficiently evaluate the conditions of tracks, including rails.

In the Netherlands, axle box acceleration (ABA) has been used since the mid-1980s to detect defects such as rail squats and poor welding [13]. Compared to other methods, ABA costs less and features easy maintenance. Recently, accelerometer-based methods have been proposed to monitor rail conditions [20–22]. Lee et al. [17] suggested a mixed filtering (Kalman filter and band-pass filter) method to estimate the irregularity of rails using accelerometers mounted on wheel axles and bogies. Shafiullah et al. [18] presented a communication protocol between accelerometers installed in a train system to monitor the typical dynamic behavior of a train. The ABA method has been used to detect rail defects, including corrugations and the welding of rails, as well as insulation joints (seams) [19]. However, the ABA-based measurement method is not capable of detecting all types of squats. Extensive detection is impossible because squats are usually generated randomly, and only one squat is generated at a position. In addition, ABA signals (especially light rail squats) may not be easily detected and analyzed without appropriate equipment and signal processing.

The present study describes the results of rail defect detection tests using an ABA system developed to detect rail squats, which are a type of fatigue defect caused by rotational contact between the wheels and the rail. Rail squats, if not repaired in time, may lead to critical consequences, such as rail breaks. Previous studies on the use of ABA systems for squat detection emphasized the need to improve hardware and signal processing performance to increase accuracy in the detection of minor squat defects. In the present study, among such improvements, those focused on the direction of the vibration measurement are summarized. One way to improve sensitivity in the rail squat measurement is to use a longitudinal ABA system. Professor Zili Li at the Delft University of Technology, the Netherlands, performed rail-wheel vibration tests using an impact hammer and concluded that the resultant frequency of the transfer function was greater in the longitudinal direction than in the vertical direction; it was also greater in the longitudinal ABA signals responding

to minor squat defects [31]. Another researcher reported that squats could be detected in the frequency range of about 2000 Hz or lower [32].

In the present study, vibration data measured from domestic railways were employed, and a method to more effectively detect the features of rail squats from ABA signals with respect to the measurement direction, as described above, was proposed. In addition, an attempt was made to upgrade the existing wavelet-based detection methods proposed in previous studies. Overall, the present study aimed to implement a more efficient detection algorithm and prediction model for rail defects, which is an integral part of all status-monitoring and error-detection systems dedicated to the various components of railway infrastructure.

## 2. Track Frequency Monitoring Using ABA

Visible rail squats are detectable with the unaided eye; other minor defects can be detected using ABA systems to analyze the frequency characteristics of vibrations. Various techniques can be used to analyze the frequency content of signals, but there is no guarantee that all these methods provide accurate information. The present study employed a wavelet-based analytical approach, which is known to ensure high resolution in terms of both time and frequency. This type of wavelet-based analysis has been regarded as a suitable method for investigating localized changes and abnormal behavior in the frequency content of signals, such as structural damage and cracks [23–29,31–36]. Notably, the current status of rails at a certain time (T) and position (x) can be expressed using $H_{T},x(f)$, a nonparametric response function that can monitor over a long distance in the longitudinal direction in railways. This can be confirmed based on the wavelet power spectrum (WPS) in the measurement frequency range; the concept of a track frequency monitoring system using ABA is outlined in Figure 1.

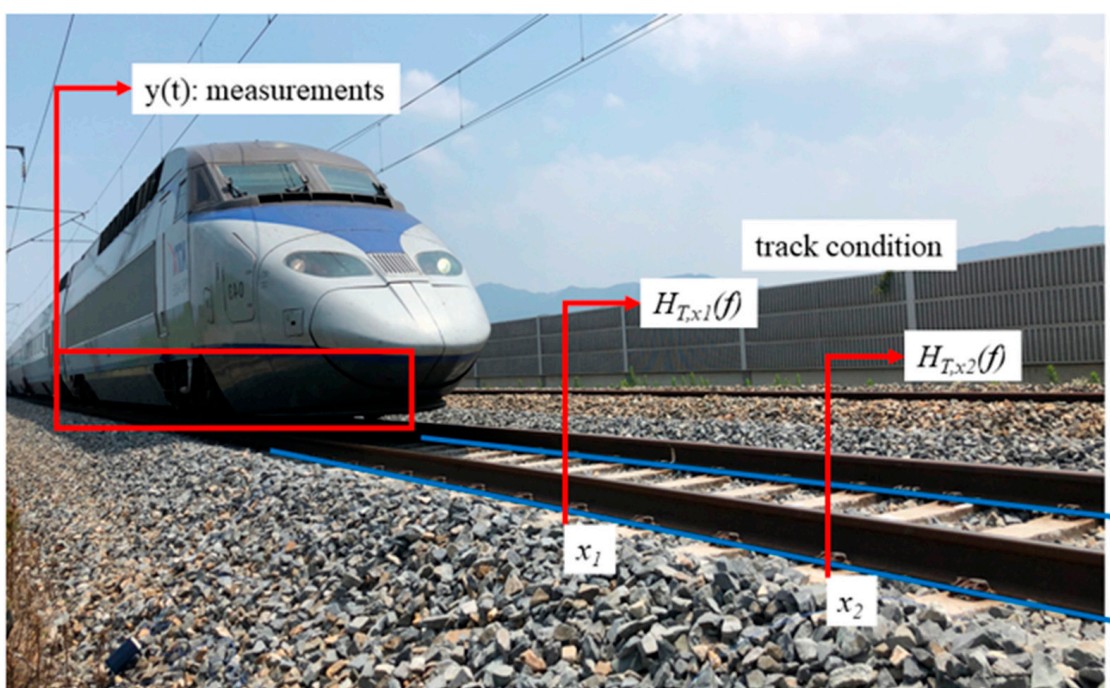

**Figure 1.** Track conditions assessed via signal-processing methods using onboard measurements.

A track frequency monitoring system using ABA can be represented by the maximum and minimum values of the WPS, depending on the response of the track. Responses vary depending on the type of track (concrete or ballast) and the type of sleeper [30,31,37–40]. In the present study, the integrity response of the track was defined using an effective $H_{T},x(f)$ model that was valid for the same track type. When a response at a specific position x differed from the normal response, the frequency (f) at which the most energy was

concentrated provided information regarding the type of irregularity near position x at time t. The position of the train x(t) was confirmed using GPS, and based on the results, the velocity of the train v(t) was estimated and then matched with vertical and longitudinal ABA measurements a(t).

$$H_T, x(f) = \left| W_n^2(s) \right| \tag{1}$$

$$W_n(s) = \sum_{n'=0}^{N-1} x_{n'} \varphi^* \left( \frac{(n'-n)\delta_t}{s} \right) \tag{2}$$

Here, $x_n$ is a time series with a time step of $\delta_t$, n is the time index, and $n' = 0, \ldots, N-1$ is the time-shift operator. $\psi$ is the mother wavelet with a locally limited function. $\psi^*$ is a wavelet inferred from the mother wavelet via different scaling. * refers to a complex conjugate, and s is the wavelet scale. The condition s > 0 holds and $W_n(s)$ is the wavelet coefficient. In the present study, a Morlet function was used as the mother wavelet.

The wavelet scale is related to the Fourier period (or inverse frequency). The wavelet conversion process can be regarded as a linear filtering operation that involves parallel filter sets. $H_T,x(f)$, a plot of the frequency response with respect to the interval $x \in X$, which has consecutive positions, is called a scalogram. The vertical slice of a scalogram for a given distance x is $H_T,x(f)$: a function that provides local spectrum measurements.

## 3. Measurement of Wheel Vibration Acceleration Using an Impact Hammer

### 3.1. Accelerometer for ABA measurement

The acceleration sensors for the ABA measurement are shown in Figure 2 and were installed one by one on the left and right sides of the wheel axle. The accelerometer is a wireless 3-axis sensor with a capacity of 60 g, and data were acquired at 6000 Hz.

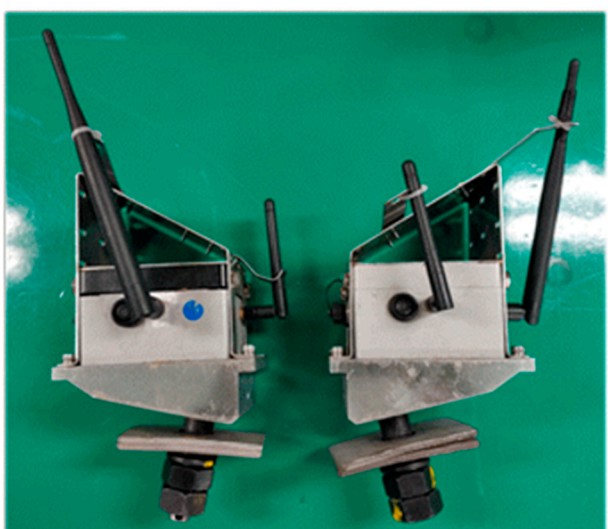

**Figure 2.** A 3-axis accelerometer.

Acceleration data were collected wirelessly from inside the vehicle, as shown in Figure 3 below, and at the same time, location information was obtained using a GPS device.

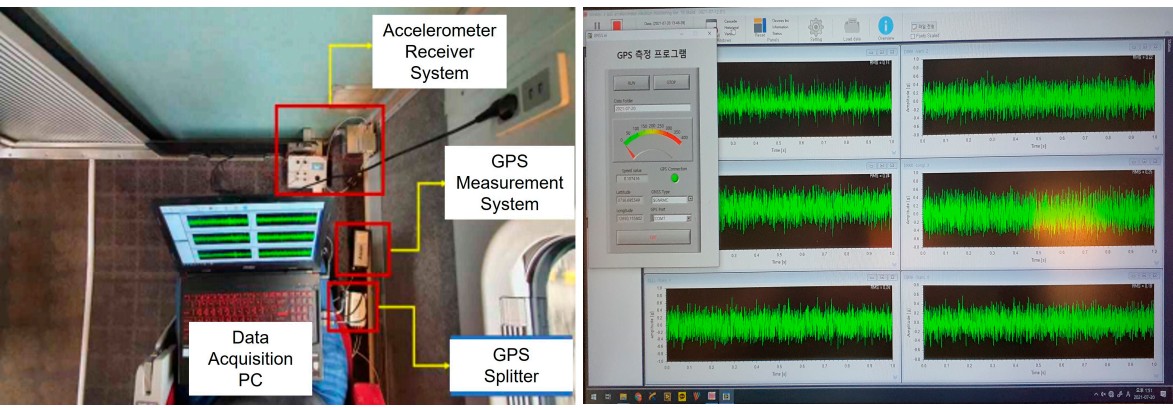

**Figure 3.** Acquisition of acceleration and GPS data.

*3.2. Design Methodology for ABA Measurement*

The purpose of using vertical ABA measurements is to detect minor squats with lengths of 10 to 30 nm and vibration on the top surface of the rail [41]. Slight deviations may occur depending on the velocity of the train of interest, but the impact of the range of frequencies induced by minor squats on forced vibration and parts is approximately 1–4 kHz [42–44]. The instrumentation setup for ABA measurement is presented in Figure 4.

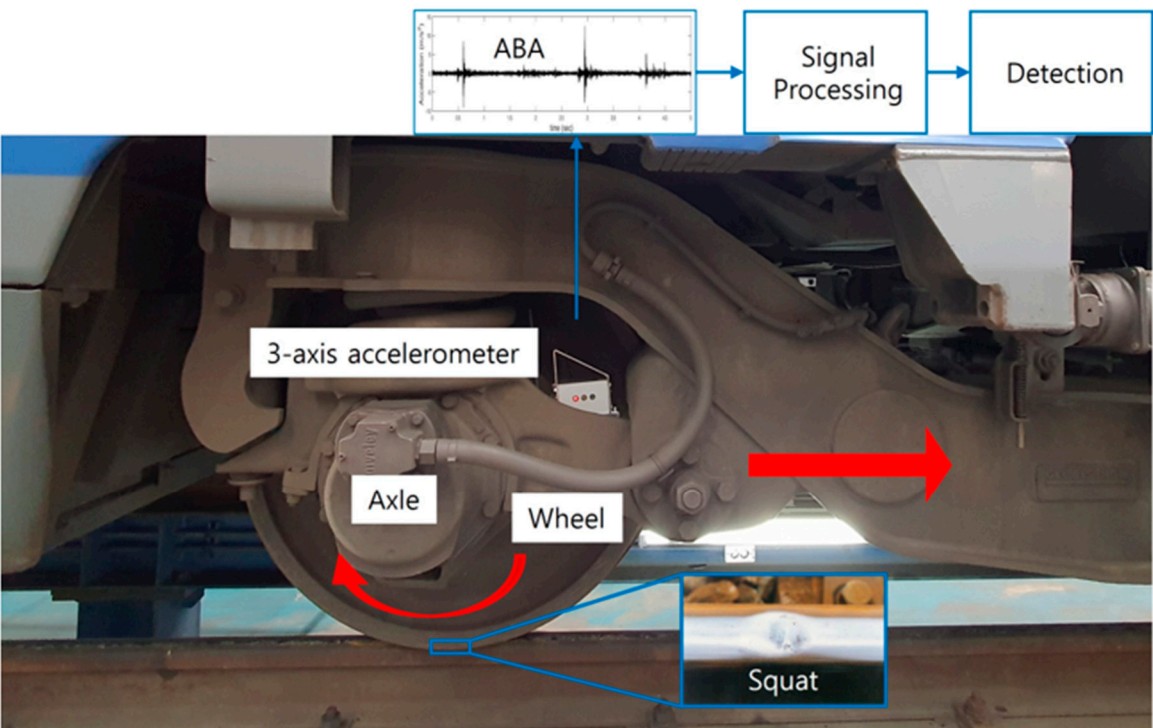

**Figure 4.** Schematic diagram and implemented prototype of instrumentation setup for ABA measurements.

Once the wheel moves over a squat, the resultant impact generates a vibration, and this vibration is then transmitted to the wheel axle. The frequency band in relation to squats can be confirmed based on ABA measurements; as a result, the presence of defects, along with the severity of squats, can be determined. In the present study, acceleration was measured using a 3-axis accelerometer. The measurement frequency was set to 6 kHz, and the maximum amplitude was 60 G. GPS coordinates were measured at a sampling frequency of 10 Hz, and the measured results were interpolated to a frequency of 6 kHz.

Meanwhile, the ABA measurement is affected by the components of the track in various ways. Different types of transfer functions are employed in the wheel-track system, as shown in Figure 5. As can be seen in the figure, short-wave defects, such as impacts, are caused by the interaction between the wheel and rail. Given that the focus of the present study was on the detection of impact-related events, any signal components other than those related to impact features were considered to be noise. In a vertical ABA system, the measured vibration included different types of vibrations occurring in bearings, wheels, rails, rail pads, fasteners, sleepers, roads, and roadbeds. By contrast, in a longitudinal ABA system, the measured vibration included only that from bearings and wheels. Accordingly, the vibration of the bearings and wheels contained in the longitudinal ABA system was almost identical to that observed in the vertical ABA system. Various types of vibration were observed in rails, rail pads, fasteners, sleepers, roads, and roadbeds and might also be caused by normal wheel–rail contact; thus, the vibration measured in the vertical ABA system was not necessarily caused by short-wave defects (squat-induced vibration). For example, vibrations resulting from the elastic deformation of the ground are in the low-frequency range and are not caused by squats. Thus, this type of vibration was considered to be noise in the present study [25,26]. However, in a practical aspect, most high-frequency vibration modes of wheels are considered to result only from wheel–rail contact, e.g., impacts [45–49]. Therefore, the present study was performed based on the idea that the longitudinal ABA system could be more sensitive to minor squats and stamping defects with less signal noise.

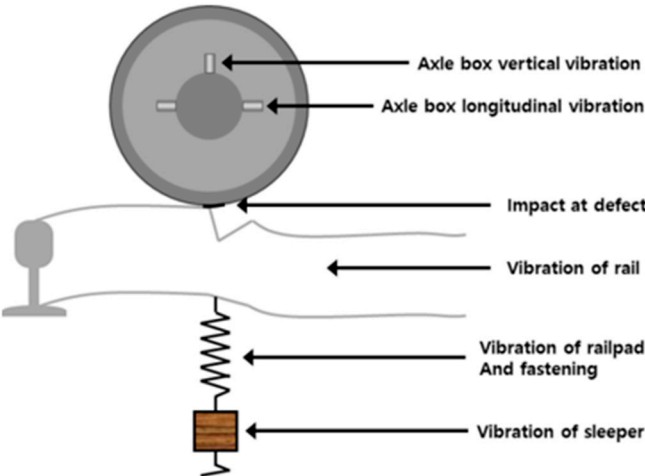

**Figure 5.** Measured vibrations of impact due to wheel–rail interaction.

### 3.3. Previous Studies on Wheel Vibration Modes

Impact hammer tests were performed on a wheel to prove the hypothesis that the signal-to-noise ratio of the longitudinal ABA system was larger than that of the vertical ABA system. A specific load was imposed on the rim of the wheel using an impact hammer, and a transfer function that accounted for the responses measured in the axis box was determined accordingly. As such, the transmission of vibrations from the wheel–rail interface to the wheel axle was analyzed.

Figure 6 presents the vibration modes of the wheel, which were determined using finite element analysis by a research team led by Professor Zilli Li at the Delft University of Technology, the Netherlands. Here, the longitudinal, vertical, and lateral directions were indicated as x, y, and z, respectively. From a theoretical perspective, a single-wheel system was assumed, and half of the wheel axle was considered. Accordingly, stress-free FE modal analysis was applied to determine the wheel's vibration mode. In Figure 6, red and blue refer to the maximum amplitudes of the vibration in the two opposing axial directions. It was also suggested that the theoretical resonant mode of the wheel was symmetric or asymmetric on the x–y plane centered on the wheel axle.

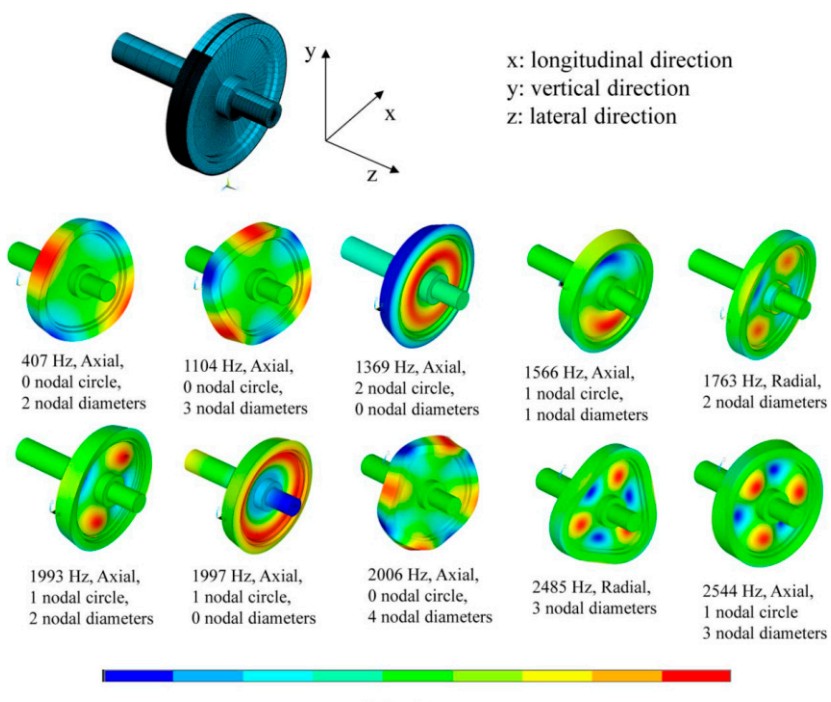

**Figure 6.** Finite-element (FE) model of wheel and wheel vibration modes; color scale indicates displacement along the axial direction [27].

### 3.4. Impact Hammer Test

In the present study, impact hammer tests were performed, as shown in Figure 7a, in which an impact was imposed on the wheel, and the resultant impact load and acceleration of the wheel axle were measured to determine the frequency response function. The impact load imposed by the impact hammer was measured using a power sensor attached to the hammer, and the resultant vibration in both vertical and longitudinal directions was measured using a 3-axis accelerometer attached to the wheel's axle. The load was applied to the rim of the wheel in the vertical direction. The results confirm that the transfer function showed higher values in the longitudinal direction than in the vertical direction in the frequency range of 820–2200 Hz, as shown in Figure 7b. The transfer function, as indicated in Figure 7b, is referred to as inertance; this value is by, definition, acceleration [m/s$^2$] divided by the force [N]. Here, the unit [m/N·s$^2$] corresponds to [1/kg]. As such, the inertance was determined by impact hammer tests, in which an impact was imposed on the wheel using the hammer five times for each set, and, subsequently, the average force F(t) and the average acceleration of the sensor a(t) were estimated. The measured average acceleration was then converted using fast Fourier transform (FFT) into a frequency range, and based on these results, inertance I(jω) was calculated as follows.

$$I(jw) = \frac{S_{\alpha F}(jw)}{S_{FF}(jw)} = \frac{\sum_{n=1}^{N} \sum_{m=1}^{N-m-1} \alpha \lceil m+n \rceil F \lceil m \rceil e^{-jwn}}{\sum_{n=1}^{N} \sum_{m=1}^{N-m-1} F \lceil m+n \rceil F \lceil m \rceil e^{-jwn}} \tag{3}$$

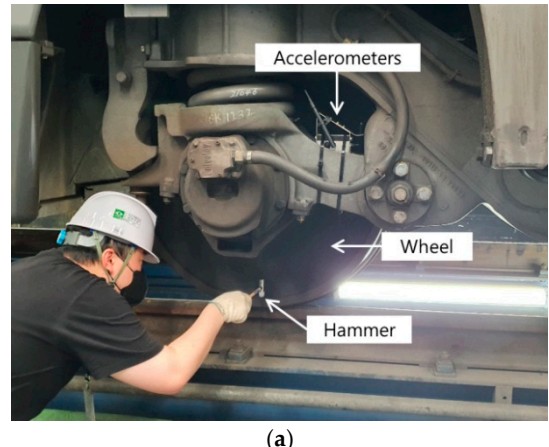
(**a**)

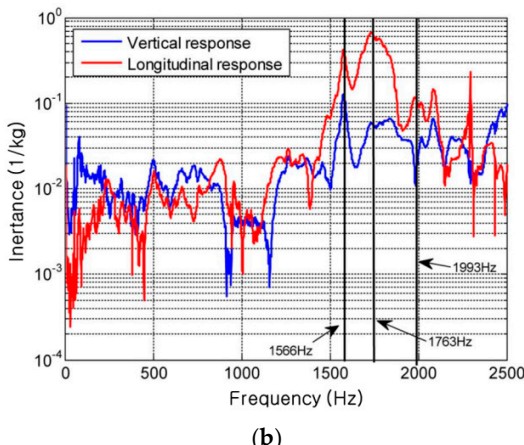
(**b**)

**Figure 7.** (**a**) Hammer test performed on the wheel and (**b**) Calculated inertance.

Here, $S_{aF}$ is the cross-spectrum between the force and acceleration, $S_{FF}$ is the auto-spectrum of the force, N is the number of sampled data points, and $\omega = 2\pi f$ (f is the frequency in Hz). The frequency range of the inertance could be defined based on the applicability of the impact hammer and measured data. Here, the range includes frequencies of 50–3000 Hz.

## 4. Analysis of Wavelet Power Spectrum

### 4.1. Comparison between Vertical and Longitudinal ABA Measurements

The vertical and longitudinal measurement results obtained using the 3-axis accelerometer attached to the wheel axle of the train were compared. As shown in Figure 8a, the intensity of the longitudinal ABA was about 1.4 times greater than that of the vertical ABA when the same rail defect was present. Notably, when the frequency characteristics were examined based on the wavelet power spectrum (WPS), this difference was more pronounced in longitudinal direction analysis, as in Figure 8b. It was accordingly expected that the application of the longitudinal ABA measurement might be even more effective at squat detection.

Here, the obtained WPS is plotted in the form of a scalogram. The vertical slice displayed on the right side of a wavelet graph presents the measure of general spectra, and its unit is dB/Hz. Examples of this type of graph are presented in Figure 8, which indicates the response of acceleration signals for squats. The closer to red the color became, the higher signal energy was concentrated within a specific frequency range, i.e., the higher intensity of the WPS. It is a relative value that varies depending on the degree of response in the entire measurement range. In the present study, the degree of response was used to define the time–frequency relationship of squats with the axle box acceleration signals.

### 4.2. Sites with Actual Squat Defects and WPS Analysis Results

For the on-site confirmation of this section where data were obtained, the location where the squat occurred was confirmed using the GPS system, as shown in Figure 9. On-site confirmation was conducted for a section of about 1 km.

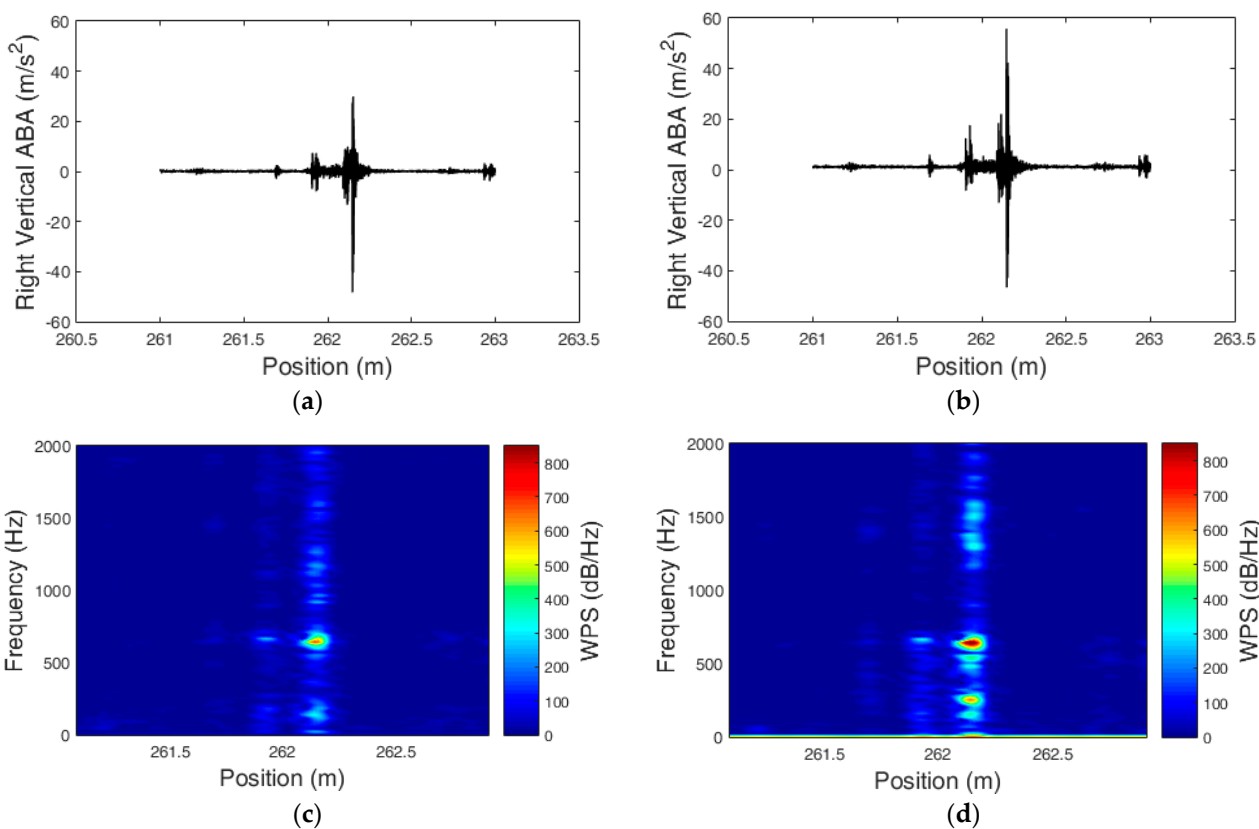

**Figure 8.** ABA measurements on a squat using the prototype. (**a**) Vertical ABA measurement. (**b**) Longitudinal ABA measurement. (**c**) WPS of vertical ABA. (**d**) WPS of longitudinal ABA.

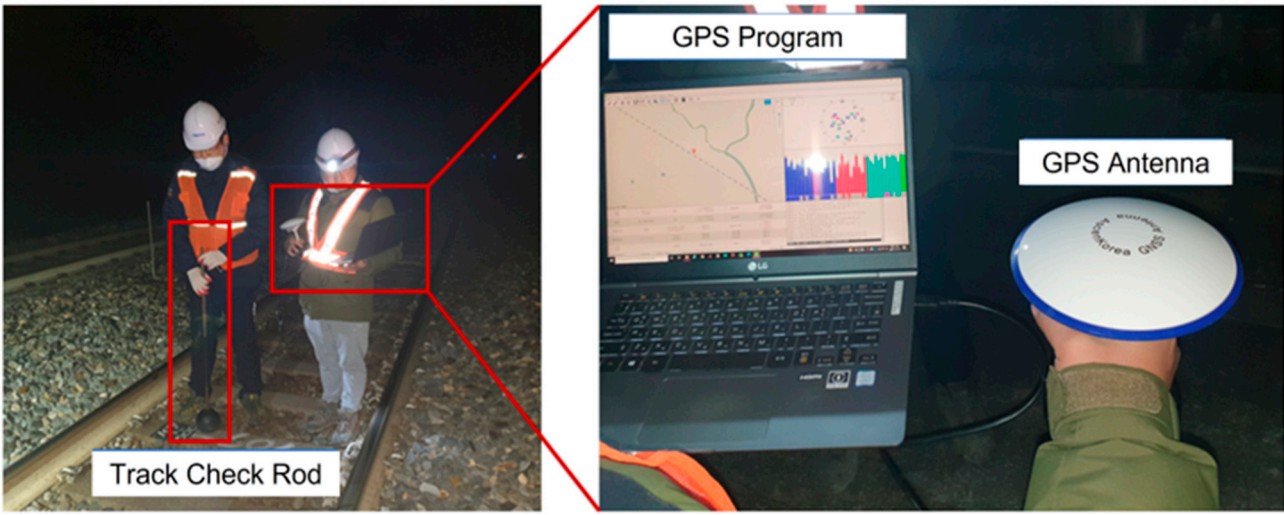

**Figure 9.** On-site confirmation of squat occurrence location.

Figure 10 presents images of actual rail squats. These two squats were found close to each other, and their presence was confirmed using vertical and longitudinal ABA measurements, as shown in Figure 11. In Figure 11, a hit means that the measurement system detected a squat, and a miss means that the measurement system misidentified it as a squat. Also, false means that the measurement system did not find the squat.

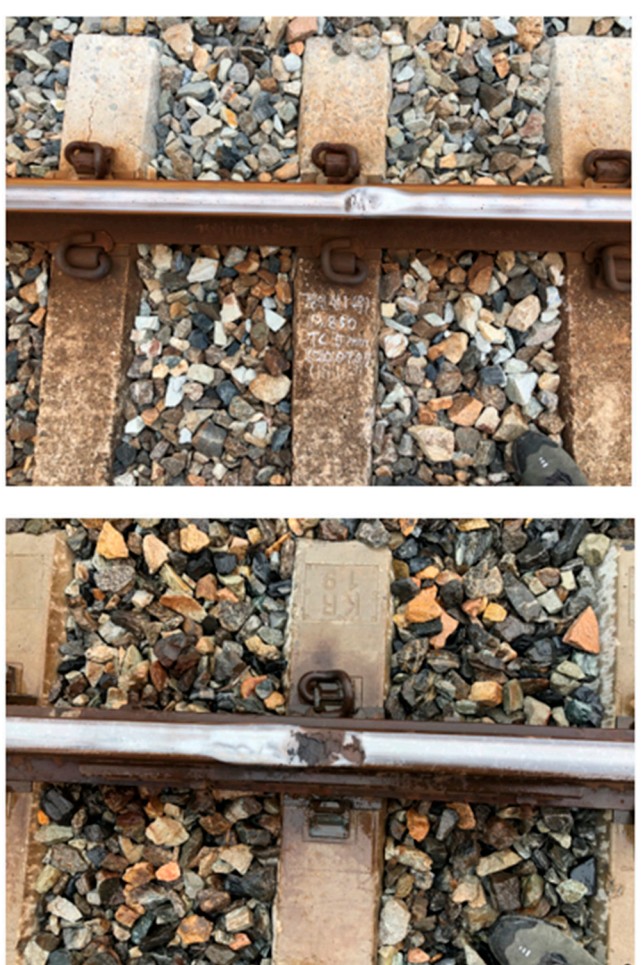

**Figure 10.** A squat on the rail.

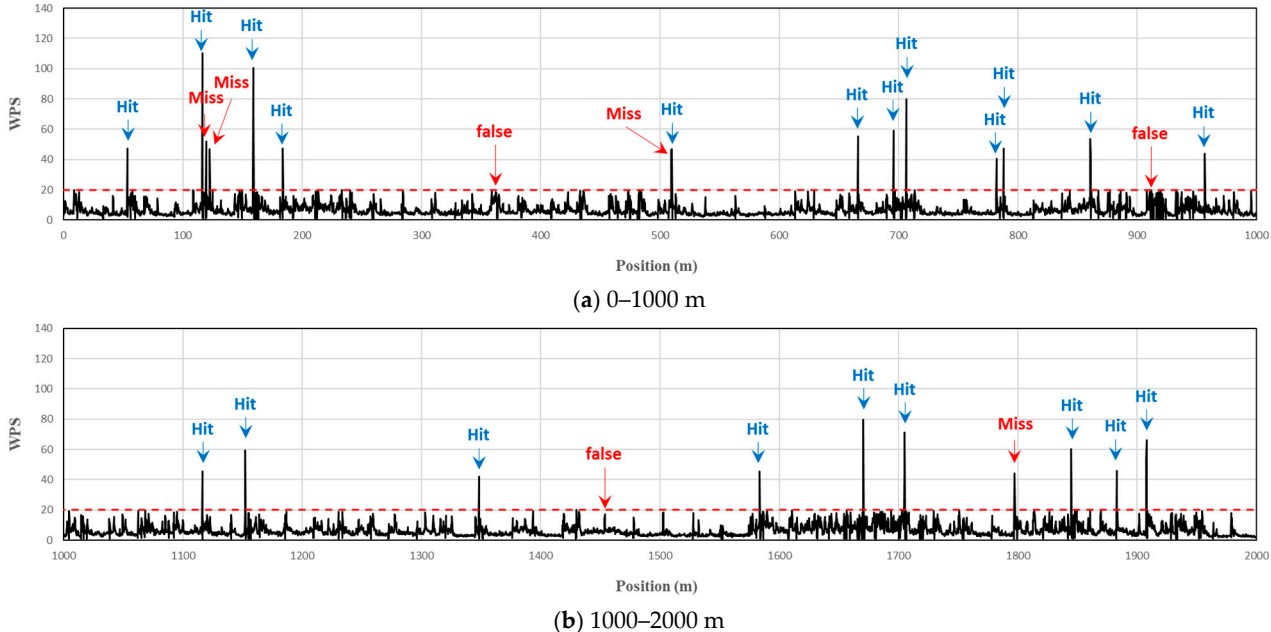

(**a**) 0–1000 m

(**b**) 1000–2000 m

**Figure 11.** *Cont.*

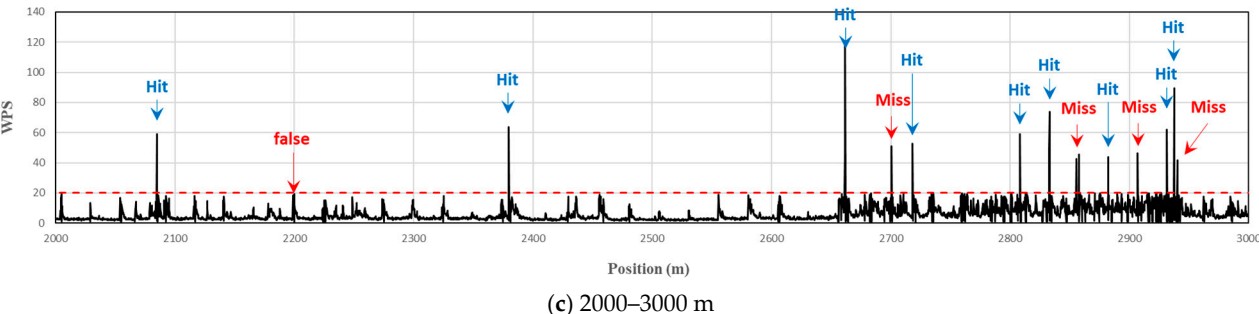

(**c**) 2000–3000 m

**Figure 11.** WPS values and detected position. (**a**) 0–1000 m (**b**) 1000–2000 m and (**c**) 2000–3000 m.

This pattern was more pronounced in the longitudinal ABA measurement than in the vertical measurement. The response to the squat defect was clearly visible in the frequency range of 600–700 Hz. The velocity within the measurement range was about 300 km/h. If this squat was present in a higher velocity range, the corresponding response could be detected at higher frequencies. In domestic railways, rail conditions are strictly managed, especially at a high-speed range, and thus, it is considered to be difficult to detect squat defects when the train speed is very high, e.g., 300 km/h. However, squats often occur in welded rail joints, and therefore, the use of an ABA system is expected to make squat detection even faster and simpler.

### 4.3. Detection of Squat Defects

The results of this study conducted in High speed railway sections showed that the outstanding frequency range of the axis' acceleration due to squats was 600 to 800 Hz, as shown in the red square in Figure 12. Therefore, WPS was high in the frequency ranges of 600 to 800 Hz. These results were applied to the squat detection program developed in the present study to predict the positions of squats. The threshold of the WPS was set to 20, and the positions showing WPS values higher than the threshold were inspected visually. A total of 34 positions of squat generation were confirmed via a visual inspection, while 38 positions were predicted as squat positions by the program developed in this study. Figure 11 shows the WPS in the measurement interval and defect positions. Table 2 shows the actual number of squats and the prediction accuracy of the developed program. With reference to 34 defect positions, the prediction accuracy was 88.2%, and the false alarm rate was 10.5%.

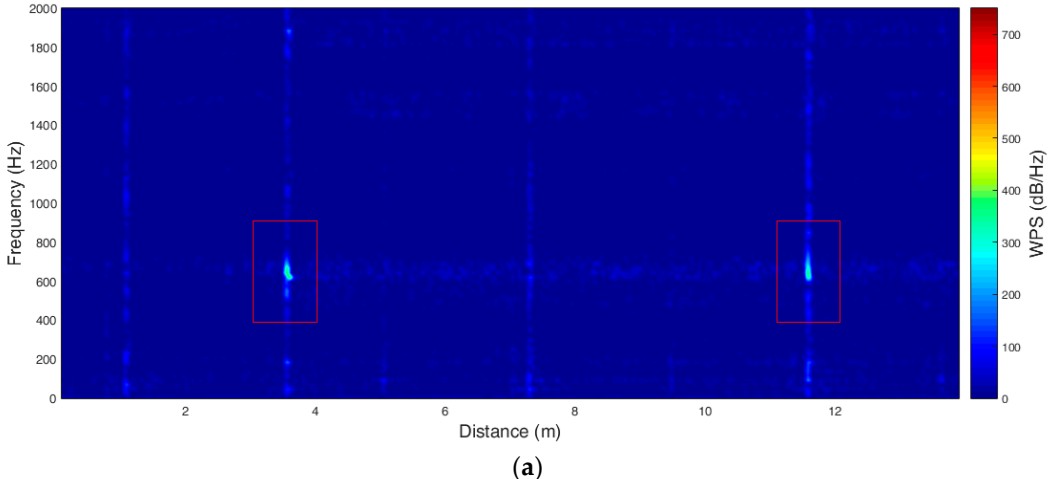

(**a**)

**Figure 12.** *Cont.*

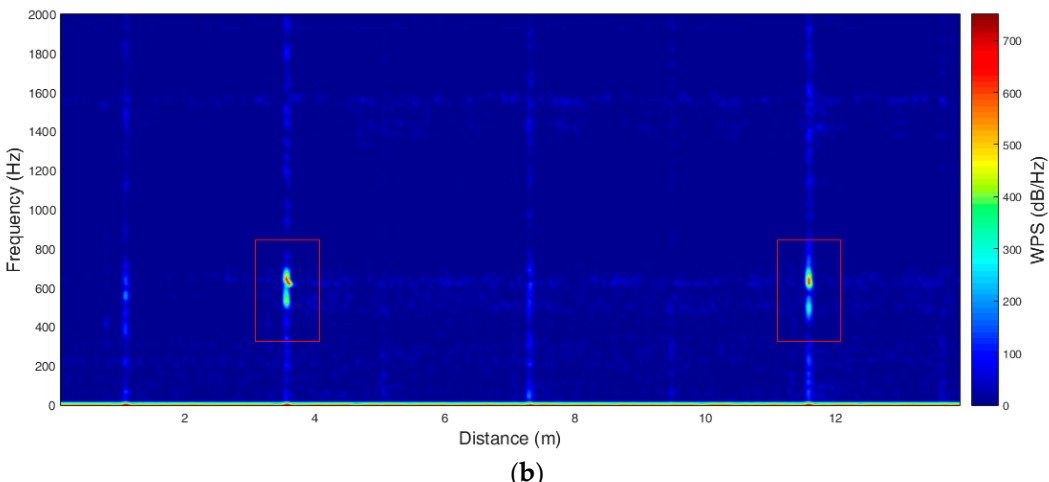

**Figure 12.** (**a**) Vertical ABA measurement. (**b**) Longitudinal ABA measurement.

**Table 2.** Squat detection rate and false alarm rate.

| Location (m) | No. of Defects | No. of Predictions | No. of Detected Defects | No. of False Alarms | HR (%) | FA (%) |
|---|---|---|---|---|---|---|
| 0~1000 | 14 | 15 | 12 | 2 | 85.7 | 13.3 |
| 1000~2000 | 10 | 10 | 9 | 1 | 90.0 | 10.0 |
| 2000~3000 | 10 | 13 | 9 | 1 | 90.0 | 7.7 |
| Total | 34 | 38 | 30 | 4 | 88.2 | 10.5 |

## 5. Conclusions

The present study attempted to verify the sensitivity measurement of a longitudinal axle box acceleration (ABA) system for squats among short-wave tracking irregularities. The obtained data were post-processed using the ABA-analysis system, and the results confirmed that longitudinal-direction vibration data were more effective than vertical-direction data in squat detection.

The proposed detection program was based on the WPS of the measurement frequency. The frequency bands associated with the squats were identified using the wavelet spectrum. The frequency bands of the squats were 600 to 800 Hz, depending on the characteristics of the rails with the squats (joints, welds, hanging sleepers, etc.).

The threshold for detecting squats on rails was established empirically. The accuracy of squat detection performed by the program was 88.2%, and the false alarm rate was 10.5%.

Given that minor rail surface defects include those even less significant than minor squats, the use of this longitudinal ABA analysis system is expected to improve accuracy for the detection of minor squat defects.

A future study aims to focus on verifying the applicability of this type of ABA system in the assessment of other components of rail infrastructure, such as insulation joints. It is worth noting that the rail misalignment and bolt fastening status cannot be observed with the unaided eye, and thus, human operators are not able to detect such defects. The analysis of rail corrugation, welded joints, turnouts, and other track system components also aims to be the focus of our future research. Not only that but further research efforts should also be made to implement this ABA system for general passenger trains so that the track network can always be monitored. The ABA system developed in the present study allows safety inspections to be performed at shorter intervals than regular inspections using special inspection vehicles, such as track inspection cars. Thus, this system can respond to the occurrence of any abnormalities in the track more promptly.

The application of the proposed measurement system with acceleration data analysis could help in efficiently performing rail maintenance and taking appropriate preventive measures. The effective maintenance method suggested in this paper could also significantly reduce the life cycle cost of rails that are affected by squats. Further studies will be conducted to improve the algorithm and develop a system that can be applied to various trains to monitor rails.

**Author Contributions:** Writing—original draft preparation, H.C.; writing—review and editing, H.C.; validation, J.P.; software, K.P. All authors have read and agreed to the published version of the manuscript.

**Funding:** This research was supported by the National Research Foundation of Korea (NRF) grant funded by the Korea government (MSIT) (No.2022M3E8A1077633).

**Data Availability Statement:** The data cannot be used due to privacy or ethical restrictions.

**Conflicts of Interest:** The authors declare no conflict of interest.

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
