# Peer review of "Analysis of Axial Acceleration for the Detection of Rail Squats in High-Speed Railways"

_2673-4109, doi:10.3390/civileng4040062_

Round 1

Reviewer 1 Report

Comments and Suggestions for Authors

The manuscript deals with the analysis of axial accelerations in high-speed railway traffic. The authors claim that, by analyzing the acceleration of axle boxes, they are able to detect rail squats. Figs 6 and 8 show that a squat causes acceleration peaks (as do other short-wave track irregularities, which has been abundantly shown in previous research). The big question, which is not answered, is: can this method discriminate between squats and other irregularities? As long as this is not proven, one cannot claim the ability of detecting squats. The experimental design needs to be enhanced and additional experiments need to be performed in order to reach this goal, and therefore the intended purpose of the manuscript.

Author Response

Thank you very much for reviewing the thesis.
Test methods and test results for rail squat measurement have been added. Please check the attachment file.

Reviewer 2 Report

Comments and Suggestions for Authors

The abstract provides a concise overview of the paper's objectives and methodology. It highlights the importance of early detection of squats for reducing maintenance costs and introduces the use of Axle Box Acceleration (ABA) and frequency signal processing techniques for detection. The abstract could be improved by mentioning the specific benefits of early detection and briefly summarizing the findings.

Comments:

The manuscript presents a study on the detection of squats in high-speed railway tracks using Axle Box Acceleration (ABA) and frequency signal processing techniques. The topic is relevant as the early detection of squats can help reduce maintenance costs and ensure safe operation. The use of ABA for longitudinal vibration measurement is interesting and has the potential to improve squat detection.

The methodology section should provide more details about the experimental setup, such as the specific parameters and configurations of the 3-axis accelerometer and the data collection process. Additionally, the signal processing techniques used for extracting squat signals should be explained in more detail. This will help readers understand the validity and reliability of the results.

The conclusion section provides a good summary of the findings, stating that longitudinal ABA data is more effective than vertical ABA data in squat detection. The suggestion of applying the longitudinal ABA system for the detection of minor squat defects and other rail infrastructure components is valuable. However, it would be beneficial to include some quantitative results or statistical analysis to support these claims.

The future research directions outlined in the conclusion are relevant and would contribute to the further development of the ABA system. However, it would be helpful to mention potential challenges or limitations that need to be addressed in future studies.

The manuscript has valuable contributions to the field of railway track maintenance and squat detection. However, some improvements are necessary to enhance the clarity and rigor of the study. Therefore, the decision is a major revision.

The specific comments are 

The abstract lacks clarity and conciseness. It would greatly benefit from clearly stating the specific findings of the study and providing more context to capture the reader's interest.

The methodology section is insufficiently detailed, making it difficult for readers to understand the experimental setup and replicate the study. Please provide more specific information about the accelerometer configuration, data collection process, and any calibration procedures conducted.

The analysis and interpretation of the results are unclear. The authors should include more detailed descriptions of the signal processing techniques used and provide clear explanations of how the squat signals were effectively extracted from the longitudinal vibration measurements.

The conclusion section is too brief and lacks substantive discussion of the implications of the findings. Please provide more comprehensive insights into the significance of the study's results and how they contribute to the field of railway track maintenance.

The manuscript would greatly benefit from the inclusion of statistical analysis or quantitative results to support the claims made throughout the paper. This would enhance the scientific rigor and credibility of the study.

The future research directions mentioned in the conclusion are vague and lack specificity. Please provide more concrete plans and objectives for future studies, including potential challenges and limitations that need to be addressed.

The language and writing style of the manuscript need significant improvement. There are numerous grammatical errors, awkward phrasings, and inconsistencies throughout the text. A thorough proofreading and revision are necessary to enhance the clarity and readability of the manuscript.

The manuscript would benefit from more thorough referencing and citation of relevant literature. Please ensure that all sources are accurately cited and that the discussion is properly grounded in existing research.

The manuscript lacks a strong motivation for the research. It would greatly benefit from a more compelling introduction that clearly establishes the importance of early squat detection and outlines the current challenges in the field.

The manuscript could benefit from the inclusion of visual aids such as figures, diagrams, or graphs to help illustrate the experimental setup, data analysis, and results. This would improve the overall clarity and facilitate understanding for readers.

Comments on the Quality of English Language

The English can be improved by professional editing.

Author Response

Thank you very much for reviewing the thesis.

  • The abstract lacks clarity and conciseness. It would greatly benefit from clearly stating the specific findings of the study and providing more context to capture the reader's interest.

 -  It was written with an abstract added.

  • The methodology section is insufficiently detailed, making it difficult for readers to understand the experimental setup and replicate the study. Please provide more specific information about the accelerometer configuration, data collection process, and any calibration procedures conducted.

 -  Added information on accelerometer configuration and data collection.

  • The analysis and interpretation of the results are unclear. The authors should include more detailed descriptions of the signal processing techniques used and provide clear explanations of how the squat signals were effectively extracted from the longitudinal vibration measurements.

 - The results were added and written.

  • The conclusion section is too brief and lacks substantive discussion of the implications of the findings. Please provide more comprehensive insights into the significance of the study's results and how they contribute to the field of railway track maintenance.

-  Added analysis and interpretation of test results.

  • The manuscript would greatly benefit from the inclusion of statistical analysis or quantitative results to support the claims made throughout the paper. This would enhance the scientific rigor and credibility of the study.

- Added information on quantitative results.

  • The future research directions mentioned in the conclusion are vague and lack specificity. Please provide more concrete plans and objectives for future studies, including potential challenges and limitations that need to be addressed.

-  A conclusion has been added.

  • The language and writing style of the manuscript need significant improvement. There are numerous grammatical errors, awkward phrasings, and inconsistencies throughout the text. A thorough proofreading and revision are necessary to enhance the clarity and readability of the manuscript.

- The manuscript has been corrected.

  • The manuscript would benefit from more thorough referencing and citation of relevant literature. Please ensure that all sources are accurately cited and that the discussion is properly grounded in existing research.

- References to related literature have been appropriately reflected.

  • The manuscript lacks a strong motivation for the research. It would greatly benefit from a more compelling introduction that clearly establishes the importance of early squat detection and outlines the current challenges in the field.

- Squats are a classic flaw in the track. It is necessary to develop technology that can reduce maintenance costs through effective inspection.

  • The manuscript could benefit from the inclusion of visual aids such as figures, diagrams, or graphs to help illustrate the experimental setup, data analysis, and results. This would improve the overall clarity and facilitate understanding for readers.

- Added figures and tables.

Reviewer 3 Report

Comments and Suggestions for Authors

The study to reduce maintenance costs by early detection of rail squats is rery interesting and actual.

The reserach is well structurerd. I suggest to add, if there are available, consiedartions about the repeatibility and errors of the proposed method.

Author Response

(The authors gave the same response as above.)

Reviewer 4 Report

Comments and Suggestions for Authors

The use of a longitudinal ABA-analysis system is expected to improve accuracy in the detection of minor squat defects. It will be useful for the inspection of rails.

Please check the comments.

Please see the pdf file.

Comments on the Quality of English Language

Fine.

Author Response

(The authors gave the same response as above.)

Round 2

Reviewer 1 Report

Comments and Suggestions for Authors

Thanks to the authors for adding experimental investigations regarding detection, misses and false detections. This is valuable information of high interest to a wider audience. The manuscript is now recommended for publication.

Reviewer 2 Report

Comments and Suggestions for Authors

All the comments from the reviewers are well addressed in revised manuscript. 

Comments on the Quality of English Language

Overall, the English writing is adequate.